# Molecular Mechanisms Driving Progression of Liver Cirrhosis towards Hepatocellular Carcinoma in Chronic Hepatitis B and C Infections: A Review

**DOI:** 10.3390/ijms20061358

**Published:** 2019-03-18

**Authors:** Tatsuo Kanda, Taichiro Goto, Yosuke Hirotsu, Mitsuhiko Moriyama, Masao Omata

**Affiliations:** 1Division of Gastroenterology and Hepatology, Department of Medicine, Nihon University School of Medicine, 30-1 Oyaguchi-kamicho, Itabashi-ku, Tokyo 173-8610, Japan; kanda2t@yahoo.co.jp (T.K.); moriyama.mitsuhiko@nihon-u.ac.jp (M.M.); 2Lung Cancer and Respiratory Disease Center, Yamanashi Central Hospital, 1-1-1 Fujimi, Kofu, Yamanashi 400-8506, Japan; 3Genome Analysis Center, Yamanashi Central Hospital, Yamanashi 400-8506, Japan; hirotsu-bdyu@ych.pref.yamanashi.jp (Y.H.); m-omata0901@ych.pref.yamanashi.jp (M.O.); 4The University of Tokyo, 7-3-1 Hongo, Bunkyo-ku, Tokyo 113-8655, Japan

**Keywords:** cirrhosis, HBV, HCV, hepatocellular carcinoma

## Abstract

Almost all patients with hepatocellular carcinoma (HCC), a major type of primary liver cancer, also have liver cirrhosis, the severity of which hampers effective treatment for HCC despite recent progress in the efficacy of anticancer drugs for advanced stages of HCC. Here, we review recent knowledge concerning the molecular mechanisms of liver cirrhosis and its progression to HCC from genetic and epigenomic points of view. Because ~70% of patients with HCC have hepatitis B virus (HBV) and/or hepatitis C virus (HCV) infection, we focused on HBV- and HCV-associated HCC. The literature suggests that genetic and epigenetic factors, such as microRNAs, play a role in liver cirrhosis and its progression to HCC, and that HBV- and HCV-encoded proteins appear to be involved in hepatocarcinogenesis. Further studies are needed to elucidate the mechanisms, including immune checkpoints and molecular targets of kinase inhibitors, associated with liver cirrhosis and its progression to HCC.

## 1. Introduction

Almost all patients with hepatocellular carcinoma (HCC), a major type of primary liver cancer, have liver cirrhosis [1], the severity of which can prevent effective treatment for HCC despite recent progress in the efficacy of anticancer drugs for advanced stages of HCC [2,3,4,5,6,7]. Because most patients with liver cirrhosis are asymptomatic, it is difficult to diagnose early stages of HCC [8], and patients with hepatic symptoms and HCC are considered to have advanced-stage HCC [8,9]. These issues explain the prevalence of poor prognosis for HCC patients.

HCC is the 4th most common neoplasm and the 2nd commonest cause of cancer deaths in the world. Notably, HCC is a male-dominant disease, with the incidence of HCC ~3-fold higher in males than in females [10]. Hepatitis B virus (HBV) infection is associated with the higher HCC incidence in persons with cirrhosis, occurring in high endemic areas and in Western countries (5-year cumulative incidence, 15% and 10%, respectively) [11]. In hepatitis C virus (HCV)-related cirrhosis, the 5-year cumulative HCC risk is 30% in Japan and 17% in Western countries [11].

Histologically, liver fibrosis involves the deposition of extracellular matrix proteins, including collagen, in higher-order structures within hepatic parenchyma [12,13], with hepatic stellate cells and fibroblasts representing major producers of collagen [14]. The histological pattern of liver fibrosis is not unique, and the extent and distribution of liver fibrosis exhibits various patterns depending on different etiologies [12,15]. Excessive liver fibrosis often develops within portal tracts and extends into the hepatic parenchyma in viral hepatitis, with these activities appearing to be associated with persistent portal inflammation [12,16]. Bridging fibrosis appears following the development of periportal fibrosis and extends across lobules to connect mesenchymal structures (portal tracts and central veins) to different extents. Generally, these processes accompany intrahepatic portosystemic vascular shunting and regeneration of hepatocytes, thereby transforming from normal hepatic architecture to nodule formation and finally establishing the structure of cirrhosis [12].

In this review, we discuss the molecular mechanisms underlying the progression of liver cirrhosis to HCC. We expect that this review will help clinicians diagnose and treat patients with liver cirrhosis and/or HCC in their daily clinical practice. Notably, ~70% of HCC patients are afflicted with HBV or HCV infection [11]; therefore, we focused on the occurrence of HCC during HBV and HCV infection (Figure 1).

## 2. Liver Cirrhosis and Its Progression to HCC with HBV Infection

### 2.1. Development of Liver Cirrhosis in Patients with Chronic Hepatitis B Infection

HBV infection causes acute and chronic hepatitis, cirrhosis, and HCC. HBV-carrier rates are higher in African and Asian countries and globally are estimated to have resulted in 786,000 deaths in 2010, the majority of which were attributed to HCC (341,400 deaths) and cirrhosis (312,400 deaths) [17]. Annual rates of development from chronic hepatitis B to liver cirrhosis ranged from 2.1 to 6.0% [18,19,20,21], and annual rates of the development of liver cirrhosis in HBV e antigen (HBeAg)-positive or anti-HBe-positive patients were 2.4% and 1.3%, respectively [18].

HBeAg-positivity and elevated HBV DNA levels are risk factors for the development of liver cirrhosis in patients with chronic hepatitis B [19]. Sumi et al. [20] reported that progression to cirrhosis is slower in HBV genotype B than that in HBV genotype C infection. Additionally, for patients with chronic hepatitis B infection, coinfection with HCV or human immunodeficiency virus (HIV) is another risk factor for the development of liver cirrhosis [21,22].

Older age (≥55 years), male gender, chronic active hepatitis, higher alanine aminotransferase (ALT) levels, history of decompensation, ferredoxin-1-associated single-nucleotide polymorphism, HLA-DQA2 rs9276370 variants, and HLA-DQB2 rs7756516 variants are also respective risk factors for the development of liver cirrhosis [18,19,20,21,23,24,25].

### 2.2. Development of HCC in Patients with HBV-related Liver Cirrhosis

Annual rates of occurrence of HCC in patients with HBV-related cirrhosis are ~2.3% [1]. Higher HBV DNA levels, HBeAg-positivity, higher HBV surface antigen (HBsAg) levels, HBV genotype C, and basal core-promoter mutations are viral risk factors for the occurrence of HCC, and older age, male gender, chronic active hepatitis, higher ALT levels, and higher α-fetoprotein levels are non-viral risk factors for the occurrence of HCC [20,23,26]. Treatment with nucleos(t)ide analogs for HBV control decreases the occurrence rates of HCC in patients with HBV-related cirrhosis [27].

## 3. Liver Cirrhosis and Its Progression to HCC with HCV Infection

HCV infection causes acute and chronic hepatitis, cirrhosis, and HCC. In 2015, there were 170,000 new HCV infections [28], with annual incidence rates of cirrhosis accompanying HCV infection at 1.1% [29]. Forns et al. [30] reported that chronic HCV infection displayed a high rate of progression to liver cirrhosis over a prolonged follow-up period, and that an aspartate aminotransferase (AST) value >70 IU/L was associated with cirrhosis development [odds ratio (OR): 4.22, 95% confidence interval (CI): 1.3–13.8]. Additionally, there is an association between HCV genotype 3 and steatosis, which accelerates fibrosis development over time in HCV genotype 3-infected patients [31,32]. Moreover, previous studies reported that exposure to HCV at a young age is associated with a reduced rate of fibrosis progression [33,34,35,36,37], and that liver fibrosis progression was mainly dependent on the age and duration of infection [36]. The evidences are growing that liver steatosis and diabetes mellitus are factors affecting progression to HCC in HCV-infected patients with compensated cirrhosis [11]. These comorbidities typically affecting aged patients might explain the increased malignant progression of the cirrhotic liver damage. Ongoing alcohol consumption and severe inflammation according to liver histology are also associated with liver fibrosis progression [35,36,37]. A multivariate analysis of HCV-infected patients [38] revealed that only male sex (OR: 3.17, 95% CI: 1.152–8.773) and HIV infection (OR: 6.85, 95% CI: 2.93–16.005) were associated with advanced liver fibrosis. Furthermore, HBsAg-positive HCV-coinfected patients are at high risk of developing liver disease [39], and injected-drug users [40]; patients with insulin resistance or diabetes [41]; and patients with concurrent obesity, diabetes, and steatosis [42] are also at risk of advanced liver fibrosis. Following antiviral treatment, sustained virological response (SVR) can reduce liver fibrosis progression in most patients infected with HCV [43].

Annual rates of HCC occurrence in patients with HCV-related cirrhosis are ~4.5% [1]. Caporaso et al. [44] found that mean age and male/female ratio were significantly higher in patients with HCC plus liver cirrhosis than in those with liver cirrhosis alone. Additionally, HBsAg-positive HCV-coinfected patients with liver cirrhosis are at a high risk of developing HCC [45], with elevated ALT levels [46] and hepatic steatosis [47] risk factors for the development of HCC in patients with HCV-related liver cirrhosis. SVR can reduce HCC development in most patients infected with HCV-related liver cirrhosis [48], although changes in the risk of HCC development following HCV eradication with direct-acting antivirals (DAAs) is controversial [49,50,51]. Further studies will be needed in DAA-era.

## 4. Molecular Mechanisms of Liver Cirrhosis and Its Progression to HCC

Accumulation of genetic mutations occurs during HCC progression. Recent advances in next-generation sequencing technology to augment Sanger sequencing enabled whole-genome sequencing to allow critical insights into the molecular mechanisms associated with this activity.

### 4.1. Driver-Gene Candidates in HCC

Totoki et al. [52] found 30 candidate driver genes [telomerase reverse transcriptase (TERT), catenin β1 (CTNNB1), tumor protein p53 (TP53), AT-rich interaction domain 2 (ARID2), axin 1 (AXIN1), TSC complex subunit 2 (TSC2), retinoblastoma protein 1 (RB1), activin A receptor type 2A (ACVR2A), bromodomain containing 7 (BRD7), cyclin dependent kinase inhibitor (CDKN)1A, menin 1 (MEN1), polypeptide N-acetylgalactosaminyltransferase 11 (GALN11), fibroblast growth factor 19 (FGF19), cyclin (CCN)D1, AT-rich interaction domain 1A (ARID1A), CDKN2A, CDKN2B, ribosomal protein S6 kinase, 90 kDa, polypeptide 3 (RPS6KA3), nuclear factor, erythroid 2 like 2 (NFE2L2), nuclear receptor corepressor 1 (NCOR1), alcohol dehydrogenase 1B, β polypeptide (ADH1B), Snf2-related CREB binding protein (CREBBP) activator protein (SRCAP), Fc receptor like 1 (FCRL1), phosphatase and tensin homolog (PTEN), heterogeneous nuclear ribonucleoprotein A2/B1 (HNRNPA2B1), cytochrome P450 family 2 subfamily E member 1 (CYP2E1), mitogen-activated protein kinase 3 (MAP2K3), tuberous sclerosis (TSC)1, transmembrane protein 99 (TMEM99), and glucose-6-phosphatase, catalytic subunit (G6PC)] and 11 core pathway modules [β-catenin, chromatin remodeling, DNA repair, estrogen-related receptor beta (ERRB), fibrinogen, mechanistic target of rapamycin kinase (mTOR), synaptic connection, TERT, p53, NOTCH, and NCOR] through the collection of data from 503 HCC genomes from different populations (Table 1).

Fujimoto et al. [53] identified 15 significantly mutated genes, including TP53, ERBB-receptor feedback inhibitor 1, Zic family member 3, CTNNB1, glucoside xylosyltransferase 1, otopetrin 1, albumin (ALB), ATM serine/threonine kinase (ATM), zinc finger protein 226, ubiquitin specific peptidase (USP)25, WW-domain-containing E3 ubiquitin protein ligase 1, immunoglobulin superfamily member 10, ARID1A, ubiquitin protein ligase E3 component n-recognin 3, and bromodomain adjacent to zinc finger domain 2B, after sequencing and analyzing the whole genomes of 27 HCCs, including 25 with HBV- or HCV-associated HCC. Additionally, whole-genome sequencing analyses of HCCs demonstrated the influence of etiological backgrounds on mutation patterns and recurrent mutations in chromatin regulators [53]. The authors found that multiple chromatin regulators, including ARID1A, ARID1B, ARID2, lysine methyltransferase 2A, and lysine methyltransferase 2C, were mutated in ~50% of the tumors, with clonal integration of the HBV genome in the TERT gene frequently detected.

### 4.2. The p53-RB Pathway

TP53 mutations have been reported in HCC in Japan and aflatoxin B1-induced HCC [59,60]. In HCC, the p53-RB pathway is altered in 72% of cases, with 68% of these a result of significantly altered genes [52]. Somatic mutations and copy number alterations in TP53, which plays a central role in apoptosis [61], are present in 31% and 37% of cases, respectively. Among the upstream genes targeted by p53, somatic mutations and copy number alterations in ATM are present in 4% and 6% of cases, respectively; those in RPS6KA3 are present in 4% and 5% of cases, respectively, and those in CDKN2A are present in 2% and 20% of cases, respectively. Among downstream genes targeted by p53, somatic mutations and copy number alterations in CDKN1A are present in 1% and 1% of cases, respectively; those in FBXW7 are present in <1% and 16% of cases, respectively, and those in CCNE1, which suppress RB1 transcription and promote cell cycle progression, are present in 0% and 2% of cases, respectively. Among upstream genes targeted by RB, somatic mutations and copy number alterations in CCND1 are present in 0% and 8% of cases, respectively. Somatic mutations and copy number alterations in RB1, which inhibit cell cycle progression, are present in 4% and 20% of cases, respectively. Additionally, RB controls the levels of p21, which is associated with hepatocarcinogenesis [62]. Overall, 72% of HCC cases harbor alterations in component genes of either the p53 or RB pathway alone or the combined p53-RB pathway (Figure 2) [52].

### 4.3. The β-catenin Pathway (WNT Pathway)

In HCC, the β-catenin pathway is altered in 66% of cases, with 51% of these involving significantly altered genes [52]. Somatic mutations and copy number alterations in CTNNB1, which induce the expression of WNT target genes and transcription of CCND1 resulting in inhibited RB1 expression, are present in 31% and <1% of cases, respectively [52]. Among the upstream genes targeted by β-catenin, somatic mutations and copy number alterations in NCOR1 are present in 2% and 29% of cases, respectively; those in FGF19 are present in 0% and 6% of cases, respectively; those in AXIN1 are present in 6% and 15% of cases, respectively; and those in APC are present in 2% and 4% of cases, respectively [52]. Overall, 66% of HCC cases harbored WNT-pathway-related alterations.

### 4.4. Chromatin and Transcription Modulators

In HCC, chromatin-remodeling pathways are altered in 67% of cases, with 41% of these involving significantly altered genes [52]. Among upstream genes associated with the nucleosome-remodeling pathway, somatic mutations and copy number alterations in ARID1B are present in <1% and 15% of cases, respectively; those in ARID1A are present in 8% and 17% of cases, respectively; those in BRD7 are present in 2% and 16% of cases, respectively; those in ARID2 are present in 10% and 2% of cases, respectively; those in switch/sucrose non-fermentable (SWI/SNF)-related, matrix associated, actin-dependent regulator of chromatin (SMARC) subfamily C member 1 are present in <1% and 4% of cases, respectively; those in SMARC subfamily B member 1 are present in <1% and 6% of cases, respectively; those in SMARC subfamily A member 4 are present in 1% and 10%, respectively; those in SMARC subfamily E member 1 are present in <1% and 3% of cases, respectively; and those in SMARC subfamily A member 2 are present in 2% and 10% of cases, respectively [52]. Additionally, Li et al. reported that 18.2% of patients with HCV-associated HCC in the United States and Europe harbored ARID2-inactivation mutations [57]. Moreover, another study reported ARID1A mutation in 14 of 110 (13%) HBV-associated HCC specimens [54]. ARID1A functions as the epigenetic regulation of hepatic lipid homeostasis, and its suppression leads to nonalcoholic fatty liver disease and nonalcoholic steatohepatitis (NASH) [58].

Among upstream genes associated with the histone-modification pathway, somatic mutations and copy number alterations in NCOR1 are present in 2% and 29% of cases, respectively; those in SRCAP are present in 3% and 10% of cases, respectively; those in SET domain bifurcated histone lysine methyltransferase 1 (SETDB1) are present in 2% and <1% of cases, respectively; and those in lysine demethylase 6A (KDM6A) are present in 1% and 5% of cases, respectively [52]. Notably, the histone methyltransferase SETDB1 promotes HCC metastasis [63,64], and KDM6A is associated with the epithelial–mesenchymal transition in HCC [65].

### 4.5. Other Pathways

In HCC, the phosphoinositide 3-kinase (PI3K)–mTOR pathway is altered in 45% of cases, with 26% of these involving significantly altered genes [52]. Somatic mutations and copy number alterations of PI3KCA (encoding the p110α subunit of PI3K) are present in 1% and <1% of cases, respectively; those in neurofibromin 1 are present in 4% and 3% of cases, respectively; those in PTEN are present in 1% and 10% of cases, respectively; those in inositol polyphosphate-4-phosphatase, type IIB, are present in 1% and 16% of cases, respectively; those in serine/threonine kinase 11 are present in <1% and 11% of cases, respectively; those in TSC1 are present in 2% and 8% of cases, respectively; and those in TSC2 are present in 5% and 14% of cases, respectively [52]. TSC1 and TSC2 suppress mTOR expression, which promotes HCC proliferation and survival. In sorafenib-treated patients with HCC, oncogenic PI3K–mTOR-pathway alterations are associated with lower disease-control rates and decreased median progression-free survival and overall survival [66].

In HCC, the nuclear factor (erythroid-derived 2)-like 2 (NRF2)–kelch-like ECH-associated protein 1 (KEAP1) pathway is altered in 19% of cases, with 5% of these involving significantly altered genes [52]. The NRF2–KEAP1 pathway is associated with an oxidative-stress response, and persistent activation of NRF2 through the accumulation of p62 is involved in HCC development [67,68,69].

There are many protein-coding genes recurrently mutated at a frequency of <5% in HCC [55,70], whereas HBV integrations and frequent noncoding mutations in the TERT promoter represent prominent examples of noncoding mutations in HCC [52,53,54,55]. Nault et al. [56] identified frequent somatic mutations resulting in the activation of the TERT promoter in HCC (59%), cirrhotic preneoplastic macronodules (25%), and hepatocellular adenomas with malignant transformation in HCC (44%). Moreover, TERT-promoter mutation represents the most frequent genetic alteration in HCC arising from the cirrhotic or non-cirrhotic liver [56], resulting in reportedly enhanced TERT activity in HCC [71].

## 5. Molecular Mechanisms of HBV-Associated HCC

### 5.1. HBV Genome Integration Promotes HCC

The discovery that HBV integrates into host chromosomes calls into question whether HBV genome integration interacts with the activation of oncogenic processes in HCC [72]. Tokino et al. [72] could not detect specific HBV genome-integration sites in chromosomes. About half of HBV DNA-cell DNA junctions are located within the so-called cohesive end region that lies between two 11-bp direct repeats (DR1 and DR2) in the HBV genome where transcription and replication of the genome are initiated. All of the integrated HBV genomes were defective in at least one site around the cohesive end region, particularly within the HBx gene [73]. Additionally, Imazeki et al. [74] detected integrated HBV DNA in all nine HBsAg-seropositive HCCs and three of 25 (12%) HBsAg-seronegative HCCs in Japan. An analysis of breakpoints within the integration region showed that 40% of breakpoints were near the 1800th nucleotide of the HBV genome, which contains an enhancer, an HBx gene, and core promoters of the HBV genome [75].

TERT is located on chromosome 5p, which is one of the most common targets for amplification in non-small cell lung cancer [76] and is reportedly directly associated with HBV genome integration [77]. Recent data from whole-genome sequencing supports this observation and confirmed the frequent observation of HBV genome integration in the TERT locus in a high clonal proportion [52,53,54,55,56].

HBx expression might play a role in hepatocarcinogenesis by interfering with telomerase activity during hepatocyte proliferation [78], which is supported by breakpoint analysis of the HBV genome [75]. HBx functions as a transcriptional activator and suppressor and has effects on hepatocellular apoptosis [79,80]. HBx proteins may upregulate the transcriptional activation of human telomerase transcriptase [81]. Cis-activation of human TERT mRNA by HBx gene may also be the mechanism in hepatocarcinogenesis [82].

Therefore, HBV genome integration into host genomes is a mechanism involved in HCC associated with HBV infection. Moreover, whole-genome sequencing demonstrated that the integration of HBV DNA into the host hepatocyte genome can be detected in 80% to 90% of HCC and in ~30% of non-HCC liver tissue adjacent to HCC [83], and that this integration appears prior to the occurrence of HCC [75]. Thus, it is possible that occult HBV infection may also accelerate hepatocarcinogenesis in HBsAg-negative patients to some extent [84].

### 5.2. Inflammation Promotes HBV-Associated HCC

In general, a high number of HBV-DNA integrations randomly distributed among chromosomes has been detected in HBV-infected liver [85]. Chronic HBV infection progresses through multiple phases, including immune tolerance, immune activation, immune control, and, in a subset of patients who achieve immune control and immune reactivation [86]. Immune-mediated liver injury is often associated with elevated ALT levels, and elevations in tumor necrosis factor-α (TNF-α) and interleukin (IL)-1β levels are often observed in the sera of HBV-infected patients [87]. Additionally, long-lasting hepatic inflammation caused by host immune responses during chronic HBV infection can promote liver fibrosis, cirrhosis and HCC progression due to accelerated hepatocyte turnover rates and the accumulation of mutations [88].

### 5.3. Epigenetic Mechanisms Involved in HBV-Associated HCC

Epigenetic mechanisms play a role in HBV-associated hepatocarcinogenesis. MicroRNAs (miRs) are endogenous noncoding RNAs (18–22 nucleotides in length) that posttranscriptionally regulate the expression of target genes. miRs bind to the 3′-untranslated region (UTR) of target mRNA, thereby inhibiting translation [89].

Several miRNAs involved in the Toll-like receptor (TLR) signaling pathway play a critical role in innate immunity against HBV infection [89]. For example, miR-145 and miR-148a target TLR3, miR-200b, miR-200c, miR148a, miR-455, and let-7-family members target TLR4, let-7b and miR-155 target TLR7, and miR-148a targets TLR9, and all of which are involved in HBV infection [89,90]. HBV might influence miR expression and induce inflammatory cytokine production [91]. Previous reports indicated that some hepatic miRs are involved in the pathogenesis of HBV-associated HCC [92,93,94,95,96,97,98,99,100,101,102,103,104,105,106,107,108,109,110,111,112,113,114,115,116,117,118,119,120,121,122,123,124,125,126,127,128,129,130,131,132,133,134,135,136,137,138,139,140,141,142], with certain serum miRs involved in HBV-associated HCC also potentially useful as biomarkers (Table 2).

Other noncoding RNAs, such as long noncoding RNAs (lncRNAs) and circular RNA (circRNA), are also involved in the pathogenesis of HBV-associated HCC [143,144,145]. Functional studies reveal that lncRNAs (100–300 kb) contribute to the onset and progression of HBV-related HCC [143], and circRNAs, which form covalently closed continuous loops by means of unique non-canonical ‘head-to-tail’ splicing in the absence of free 3′ or 5′ ends, might promote the development of HBV-associated HCC [144,145].

Epigenetic silencing of genes, such as methylation of promoter regions, also regulates gene expression in HBV-associated HCC and could potentially serve as a diagnostic and prognostic marker of the disease [146,147]. Additionally, HBV can cause epigenetic changes by altering the methylation state of cellular DNA, the posttranslational modification of histones, and miR expression [146,148], all of which are also critical for the pathogenesis of HBV-associated HCC.

### 5.4. Roles of HBV-encoded Proteins

HBV is a partially double-stranded DNA virus (genome length: 3200 bp) and a member of the Orthohepadnavirus genus and the Hepadnaviridae family [149]. The HBV genome encodes at least four proteins [HBsAg, a core protein (splice variant: HBeAg), a DNA polymerase, and the HBx protein] that are translated from mRNAs transcribed from HBV covalently closed circular DNA and/or from HBV genome sequences integrated into the host genome [150]. Viral protein translation is initiated through binding of the PreS2 domain of HBV surface antigens to protein kinase C (PKC)α/β, which triggers PKC-dependent activation of Raf-1 proto-oncogene serine/threonine kinase (RAF-1)/MAP2K signaling and transcription factors, such as activator protein-1 (AP-1) and nuclear factor kappaB (NF-kB), resulting in the increased proliferation of hepatocytes [151]. PreS2-mediated activity subsequently upregulates the expression of the transcriptional coactivator with PDZ-binding motif by repressing miRNA-338-3p, which promotes HCC proliferation and migration [152]. Moreover, a previous study showed that a truncated mutant of HBsAg increases HBV-related tumorigenesis in a mechanism potentially associated with the downregulated expression of tumor growth factor (TGF)BI associated with the TGFβ–SMAD pathway [153]. Furthermore, HBsAg enhances the IL-6–STAT3 pathway, thereby increasing the HBsAg-mediated malignant potential of HBV-associated HCC [154].

The HBV core protein enhances cytokine production [155], and HBeAg is reportedly associated with the host immune response and cytokine production [156,157,158], both of which play roles in HBV-associated HCC.

Numerous studies reported an association between the HBx protein and hepatocarcinogenesis [159,160,161,162,163,164,165,166,167,168,169,170]. Integrated HBV DNA harbors the conserved sequences for genes encoding the core protein, surface antigens, and HBx along with their respective promoter sequences [159,160], suggesting that HBx is important for hepatocarcinogenesis. HBx represents a transactivating factor [161], and transgenic mice expressing HBx develop HCC [162]. HBx transactivates binding sites for the transcription factors AP-1 and NF-κB [163], activates the p53-RB [164,165] and β-catenin [164,165,166,167,168,169,170] pathways, and is involved in chromatin remodeling [171,172,173] and transcriptional modulation in hepatocarcinogenesis [174].

Overexpression of the HBV polymerase due to core-gene deletion enhances HCC-cell growth by inhibiting miR-100 [175]. A previous study showed that transgenic mice expressing the reverse-transcriptase domain of HBV polymerase in their livers developed early cirrhosis with steatosis by 18 months of age, with 10% subsequently developing HCC [176].

## 6. Molecular Mechanisms of HCV-Associated HCC

### 6.1. Inflammation Promotes HCV-Associated HCC

Cytokines reflect the degree of inflammation in the liver of patients with chronic hepatitis C, with this production possibly related to HCC development [177]. Takano et al. [1] prospectively investigated the incidence of HCC in 124 cases with HCV infection, finding that HCC occurred in 13 cases that included 12 cirrhotic livers and only 1 non-cirrhotic liver and suggesting that most HCC occurs in advanced liver diseases in HCV-infected individuals [178]. Because hepatic inflammation plays a role in the development of advanced liver diseases, inflammation is an important aspect in the development of HCC associated with chronic HCV infection.

### 6.2. Epigenetic Mechanisms Involved in HCV-Associated HCC

Epigenetic mechanisms also play a role in HCV-associated hepatocarcinogenesis. Previous studies reported the involvement of specific hepatic and serum miRs in the pathogenesis of HCV-associated HCC [179,180,181,182,183,184,185,186,187,188,189,190,191,192,193,194,195,196], with some of the serum miRs representing potentially useful biomarkers of the disease (Table 3). Notably, the number of reports of miRs involved in HCV-associated HCC is smaller than that related to HBV-associated HCC (Table 2).

Previous studies reported that lncRNAs are involved in the pathogenesis of HBV-associated HCC [143,144,145], with functional studies revealing that these lncRNAs contribute also to the onset and progression of HCV-related HCC [195,197]. Zhang et al. [198] reported LINC01419 transcripts expressed at higher levels in early stage HCC as compared with levels observed in dysplastic tissue. Moreover, this study also reported increased and decreased levels of AK021443 and AF070632 in advanced HCC, respectively, and that LINC01419 and AK021443 regulated the expression of genes associated with cell cycle progression, whereas AF070632 was associated with cofactor binding, oxidation–reduction activity and carboxylic acid catabolic processes [198].

Upregulated lncRNAs associated with urothelial carcinoma associated-1 (lncRNA-UCA1) and WD-repeat-containing antisense to TP53 (lncRNA-Wrap53) potentially serve as novel serum biomarkers for HCC diagnosis and prognosis [199]. Additionally, the lncRNA associated with activated by TGFβ (lncRNA-ATB) is a key regulator of TGFβ signaling and is positively correlated with the development of liver cirrhosis and HCC-specific vascular invasion [200]. Moreover, lncRNA-ATB might represent a novel diagnostic biomarker and potential therapeutic target for HCV-related hepatic fibrosis [200]. A previous study showed that HCV infection upregulates the expression of miR-373 and Wee1-like protein kinase (WEE1), a pivotal player in the G2/M transition in the cell cycle (although WEE1 is a direct target of miR-373). This study also showed that miR-373 forms a complex with the LINC00657, resulting in release of their common target, WEE1, in HCV-infected cells, and the promotion of uncontrolled cell growth [195]. Another study reported that hypermethylation of promoter regions suppressed mRNA expression, which played a role in the progression of HCV-associated HCC [201].

### 6.3. Roles of HCV-coding Proteins

We previously reported the roles for HCV-coding proteins in hepatocarcinogenesis [178], with the HCV core proteins and NS5A reportedly playing important roles in HCC development [202,203,204,205,206]. HCV-infected patients showed a higher prevalence of diabetes mellitus and insulin resistance (IR) relative to those with HBV infection [207]. IR measured using a homeostasis model assessment of IR (HOMA-IR) is significantly associated with HCC development in patients with or without chronic HCV infection. Moreover, patients with NASH, which is associated with elevated HOMA-IR, have an increased risk of liver fibrosis, cirrhosis, and HCC [208,209].

A previous study showed that the HCV core protein downregulates insulin receptor substrate (IRS)1 and IRS2 by upregulating SOCS3 levels [210]. Moreover, cross-talk between the HCV core protein and molecules regulating insulin signaling might affect HCV-associated hepatocarcinogenesis [210,211,212,213,214]. Insulin-like growth factors initiate tyrosyl phosphorylation of IRS1 and activate multiple signaling pathways essential for liver growth and HCC [215], with the activation of IRS1-mediated signaling potentially contributing to hepatic oncogenesis.

CCAAT/enhancer-binding protein (C/EBPβ) and HCV NS5A might be essential components promoting increased gluconeogenesis associated with HCV infection [216]. Previous studies reported the involvement of HCV NS5A in enhanced gluconeogenic gene expression associated with impaired insulin signaling [217,218].

Sphere-forming hepatocytes express several cancer stem-like cell (CSC) markers, including c-Kit. Previous studies showed that the HCV core protein significantly upregulates c-Kit expression at the transcriptional level, and that HCV infection potentiates CSC generation [219,220]. Additionally, a Western diet high in cholesterol and saturated fat (HCFD) in combination with the translation of HCV NS5A stimulates TLR4–Nanog homeobox (NANOG) and leptin receptor–phosphorylated STAT3 signaling, resulting in liver tumorigenesis through an exaggerated mesenchymal phenotype involving prominent Twist1-expressing tumor-initiating stem-like cells expressing NANOG [221].

## 7. HCC-specific Molecular Mechanisms from a Therapeutic Point of View

Sorafenib, regorafenib and lenvatinib are approved therapies as oral molecular-targeting drugs for advanced stages of HCC [3,4,5,6]. Sorafenib is an oral serine/threonine kinase inhibitor targeting the extracellular-signaling-regulated kinase/MAPK pathway, vascular endothelial growth factor receptor (VEGFR), platelet-derived growth factor receptor (PDGFR), and epithelial growth factor receptor (EGFR). Additionally, it acts as a tyrosine kinase inhibitor targeting VEGFR1, VEGFR2, VEGFR3, PDGFRβ, ret proto-oncogene (RET), and fms-related tyrosine kinase 3, resulting in the inhibition of tumor proliferation [222].

Regorafenib is an oral multiple kinase inhibitor of VEGFR1, VEGFR2, VEGFR3, TEK receptor tyrosine kinase, PDGFRβ, fibroblast growth factor receptor (FGFR), proto-oncogene receptor tyrosine kinase (KIT), RET, RAF-1, and B-RAF. Lenvatinib is an oral tyrosine kinase inhibitor targeting VEGFR1/2/3, FGFR1/2/3/4, PDGFRα, KIT, and RET to inhibit tumor angiogenesis and growth [222]. Therefore, sorafenib, regorafenib, and lenvatinib represent multiple kinase inhibitors that do not suppress a specific molecule, thereby restricting their use only in patients with advanced HCC and well-compensated liver function.

Nivolumab, pembrolizumab and tislelizumab are anti-programmed death-1 (PD-1) antibodies. The immune-checkpoint molecule PD-1 is a receptor that negatively regulates immune responses [222,223] via binding of its representative ligands PD-ligand (PD-L)1 and PD-L2. Inhibition of this pathway can eliminate tumors by recovering their immunosuppressive effects and restoring innate immune activity [222,224]. Cytotoxic T lymphocyte-associated antigen-4 (CTLA-4) is another counterreceptor for the B7 family of costimulatory molecules and a negative regulator of T cell activation. A blockade of the inhibitory effects of CTLA-4 is supposed to enhance immune responses against tumor cells [225] and also represents a type of immune checkpoint. The blockade of such immune checkpoints represents a promising therapeutic option for HCC; therefore, further studies are needed to determine other immune checkpoints, as well as potential targets of kinase inhibitors related to liver cirrhosis to HCC.

## 8. Conclusions

This review describes the molecular mechanisms of liver cirrhosis and its progression to HCC (Figure 3). Recent progress in next-generation sequencing has revealed several HCC-specific driver-gene candidates. Additionally, ~70% of HCC cases are caused by HBV and/or HCV infection. Although nucleos(t)ide analogs and direct-acting antivirals can control HBV and HCV replication, HCC occurrence is occasionally observed [49,50,51], and better biomarkers for early detection of HCC are needed. Furthermore, genetic and epigenetic factors, such as miRs, are involved in liver cirrhosis and its progression to HCC. Despite significant advances, additional studies are required to elucidate other molecular mechanisms, including immune checkpoints and molecular targets of kinase inhibitors associated with liver cirrhosis and its progression to HCC.

## Figures and Tables

**Figure 1 ijms-20-01358-f001:**
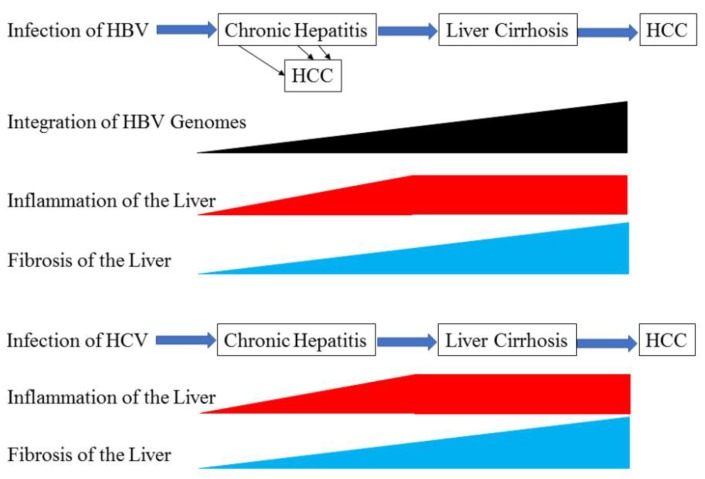
Occurrence of HCC in natural course of HBV and HCV infection.

**Figure 2 ijms-20-01358-f002:**
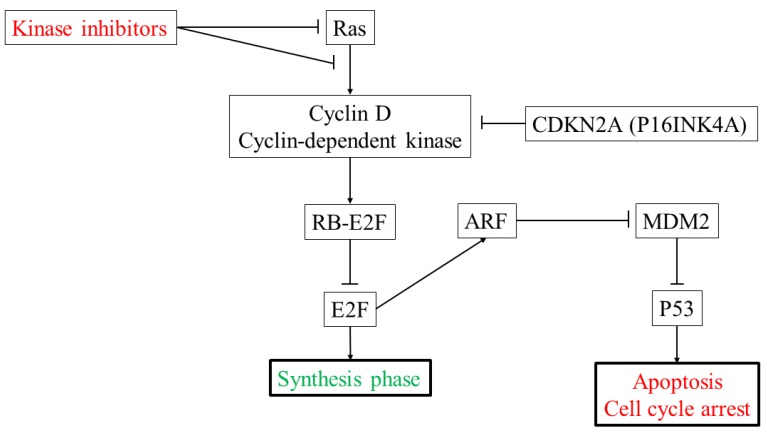
The p53-RB pathway in HCC.

**Figure 3 ijms-20-01358-f003:**
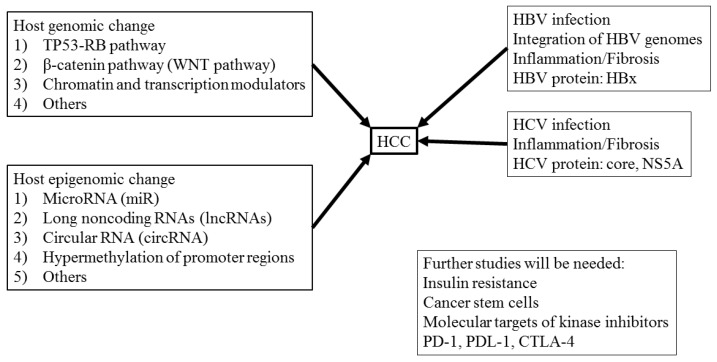
Molecular mechanisms of liver cirrhosis and its progression to HCC.

**Table 1 ijms-20-01358-t001:** Representative driver-gene candidates in HCC.

Gene Symbol	Gene Name	Pathways	References
*TERT*	Telomerase reverse transcriptase	TERT	[52,53,54,55,56]
*CTNNB1*	Catenin β1	β-catenin	[52,53]
*TP53*	Tumor protein p53	p53–RB	[52,53]
*ARID2*	AT-rich interaction domain 2	Chromatin remodeling	[52]
*AXIN1*	Axin 1	β-catenin	[52]
*TSC2*	TSC complex subunit 2	PI3K–mTOR	[52]
*RB1*	Retinoblastoma 1	p53–RB	[52]
*ACVR2A*	Activin A receptor type 2A	SMAD	[52]
*BRD7*	Bromodomain containing 7	Chromatin remodeling	[52]
*CDKN1A*	Cyclin-dependent kinase inhibitor 1A	β-catenin	[52]
*MEN1*	Menin 1	(MEN1 syndrome)	[52]
*GALN11*	Polypeptide *N*-acetylgalactosaminyltransferase 11	NOTCH	[52]
*FGF19*	Fibroblast growth factor 19	β-catenin	[52]
*CCND1*	Cyclin D1	p53–RB	[52]
*ARID1A*	AT-rich interaction domain 1A	Chromatin remodeling	[52,53,54,57,58]
*CDKN2A*	Cyclin-dependent kinase inhibitor 2A	p53–RB	[52]
*CDKN2B*	Cyclin-dependent kinase inhibitor 2B	p53–RB	[52]
*RPS6KA3*	Ribosomal protein S6 kinase, 90kDa, polypeptide 3	p53–RB	[52]
*NFE2L2*	Nuclear factor, erythroid 2-like 2	NRF2–KEAP1	[52]
*NCOR1*	Nuclear receptor corepressor 1	β-catenin/chromatin remodeling	[52]
*ADH1B*	Alcohol dehydrogenase 1B, β polypeptide		[52]
*SRCAP*	Snf2-related CREB binding protein activator protein	Chromatin remodeling	[52]
*FCRL1*	Fc receptor like 1		[52]
*PTEN*	Phosphatase and tensin homolog	PI3K–mTOR	[52]
*HNRNPA2B1*	Heterogeneous nuclear ribonucleoprotein A2/B1	MAPK	[52]
*CYP2E1*	Cytochrome P450 family 2 subfamily E member 1		[52]
*MAP2K3*	Mitogen-activated protein kinase 3	MAPK	[52]
*TSC1*	Tuberous sclerosis 1	mTOR	[52]
*TMEM99*	Transmembrane protein 99		[52]
*G6PC*	Glucose-6-phosphatase, catalytic subunit	FoxO	[52]

**Table 2 ijms-20-01358-t002:** Hepatic and serum miRs involved in HBV-associated HCC.

miRs	Upregulation/Downregulation	Target Genes	References
***Hepatic miRs***			
miR-223	Upregulation	Stathmin 1 (STMN1)	[92]
miR-143	Upregulation	Fibronectin type III domain containing 3B (FNDC3B)	[93]
miR-602	Upregulation	Ras-association domain family member 1 (RASSF1A)	[94]
miR-224	Upregulation	N/A	[98,110,116]
miR-22	Upregulation in male HCC- adjacent tissue	Estrogen receptor alpha (ERα)	[99]
miR-96	Upregulation	N/A	[112]
miR-183	Upregulation	N/A	[112]
miR-196a	Upregulation	N/A	[112]
miR-545/374a	Upregulation	Estrogen-related receptor gamma (ESRRG)	[120]
miR-331-3p	Upregulation	Inhibitor of growth family member 5 (ING5)	[124]
miR-519a	Upregulation	Forkhead box F2 (FOXF2)	[124]
miR-106b	Upregulation	N/A	[131]
miR-1269b	Upregulation	Cell division cycle 40 homolog (CDC40)	[132]
let-7a	Downregulation	Signal transducer and activator of transcription 3 (STAT3)	[95]
miR-152	Downregulation	DNA methyltransferase 1 (DNMT1), TNFRF6B	[96,115]
miR-145	Downregulation	Cullin 5 (CUL5)	[98,125]
miR-199b	Downregulation	N/A	[98]
miR-29c	Downregulation	TNF-α-induced protein 3 (TNFAIP3)	[100]
miR-92	Downregulation	N/A	[102]
miR-338-3p	Downregulation	3′ UTR region of CCND1	[104]
miR-34a	Downregulation	C-C motif chemokine ligand 22 (CCL22)	[115]
miR-101	Downregulation	DNA methyltransferase (DNMT)3A	[116]
miR-122	Downregulation	Pituitary tumor-transforming gene 1 (PTTG1) binding factor (PBF)	[117,134,139]
miR-148a	Downregulation	Hematopoietic pre-B cell leukemia transcription factor-interacting protein (HPIP)	[108]
miR-22	Downregulation	CDKN1A	[109,134]
let-7c	Downregulation	N/A	[112]
miR-138	Downregulation	N/A	[112]
miR-205	Downregulation	N/A	[113]
miR-101-3p	Downregulation	RAB GTPase 5A (RAB5A)	[114]
miR-429	Downregulation	NOTCH1	[121]
miR-216	Downregulation	Insulin-like growth factor 2 mRNA-binding protein 2 (IGF2BP2)	[122]
miR-34c	Downregulation	Transforming growth factor-β-induced factor homeobox 2 (TGIF2)	[127]
miR-18a	Downregulation	Connective tissue growth factor (CTGF)	[133]
miR-30b-5p	Downregulation	DNMT3α, USP37	[135]
miR-384	Downregulation	Pleiotrophin (PTN)	[136]
miR-125a-5p	Downregulation	erb-b2 receptor tyrosine kinase 3 (ERBB3)	[137]
miR-26a	Downregulation	N/A	[140]
miR-302c-3p	Downregulation	TNF receptor-associated factor 4 (TRAF4)	[141]
miR-1271	Downregulation	CCNA1	[142]
***Serum miRs***			
miR-25	Upregulation	N/A	[97]
miR-375	Upregulation	N/A	[97]
let-7f	Upregulation	N/A	[97]
miR-122	Upregulation	N/A	[99]
miR-18a	Upregulation	N/A	[103]
miR-101	Upregulation	N/A	[111]
miR-24-3p	Upregulation	N/A	[117]
miR-545/374a	Upregulation	ERR gamma (ERRG)	[120]
miR-96	Upregulation	N/A	[129]
miR-21	Downregulation	N/A	[118]
miR-222	Downregulation	N/A	[118]
miR-29	Downregulation	N/A	[119]
miR-150	Downregulation	N/A	[123]
miR-126	Downregulation	N/A	[128]
miR-125b	Downregulation	N/A	[130]
miR-143/145	Downregulation	N/A	[138]

**Table 3 ijms-20-01358-t003:** Hepatic and serum miRs involved in HCV-associated HCC.

miRs	Upregulation/Downregulation	Target Genes	References
***Hepatic miRs***			
miR-193b	Upregulation	STMN1	[179]
miR-155	Upregulation	N/A	[180]
miR-122	Upregulation	Cyclin G1	[181,182,186]
miR-373	Upregulation	Wee1	[195]
miR-24	Downregulation	N/A	[183]
miR-27a	Downregulation	N/A	[183]
miR-198	Downregulation	N/A	[184]
miR-152	Downregulation	N/A	[186]
miR-181c	Downregulation	Homeobox A1	[187]
miR-431	Downregulation	N/A	[189]
miR-138	Downregulation	TERT	[192]
***Serum miRs***			
miR-150	Upregulation	N/A	[191]
miR-221	Upregulation	N/A	[193]
miR-101-1	Upregulation	N/A	[193]
miR-27a	Upregulation	N/A	[194]
miR-221	Downregulation	Suppressor of cytokine signaling (SOCS)1 and SOCS3	[188]
miR-16	Downregulation	N/A	[190]
miR-34a	Downregulation	Heat-shock protein 70	[194]

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
