# Peer review of "Molecular Mechanisms Driving Progression of Liver Cirrhosis towards Hepatocellular Carcinoma in Chronic Hepatitis B and C Infections: A Review"

_ijms, 2019, doi:10.3390/ijms20061358_

Reviewer 1 Report

The review written by Kanda et al. outlines the molecular mechanisms driving progression of liver cirrhosis towards hepatocellular carcinoma (HCC) in the setting of HBV and HCV chronic infections.

Although the review provides a thorough discussion of the multiple pathways involved and their relationship with candidate driver oncogenes in HCC pathogenesis, it is not well comprehensive and some parts need improvement.

A major weakness of the manuscript is a couple of sentences that look identical to published papers that are quoted: the authors must be very careful in doing that and an accurate rewriting is mandatory.

The title is incomplete, and it should include HBV and HCV infections, since other etiologies of liver cirrhosis have not been discussed in the present review.

Furthermore, there are several specific concerns worth being addressed:

1)    Introduction. Since the paper underlines that incidence of HCC in cirrhosis related to HBV and HCV infections is higher in Japan than in Western countries, it would be important to compare epidemiology studies between the two geographical areas (see Fattovich G, Gastroenterology 2004).

2)    Row 91-93: Age-dependence of the risk to develop HCC must be clarified, taking into account that some comorbidities typically affecting aged patients (i.e. type 2 diabetes) might explain the increased malignant progression of the cirrhotic liver damage.

3)    Row 107-108: changes in the risk of HCC development following HCV eradication with DAAs is a controversial and timely issue that must be underlined (see Alberti A, Liver Int 2017).

4)    Row 143-159: mechanisms related to perturbations of p53/RB pathway are quite confused, maybe a Figure could be of help to clarify them.

5)    Row 224: the concept dealing with Coh type is unclear.

6)    Row 238-239: given the relevance of HBX in hepatocarcinogenesis, this observation should be expanded.

7)    Although the review aims at discussing the molecular mechanisms driving progression towards HCC in cirrhosis, the authors draw much attention on HBV infection. Thus, the risk of HCC in HBV occult should be commented.

8)    Iconography is limited and the sequence of Figure is not well integrated into the outlines of the main text. The reviewer suggests a swap between the two figures, moving Figure 1 to page 13 before heading #7, whilst Figure 2 would be better appreciated before the heading #4, to sum up the mechanisms involved.

Minor concern is about

1)    Row 102: it seems to be HCV-related rather than HBV.

Author Response

To Reviewer #1: Thank you for your valuable comments and criticisms.

Response to your comment:The title is incomplete, and it should include HBV and HCV infections, since other etiologies of liver cirrhosis have not been discussed in the present review.”

Thank you for your valuable comment.We agree with you. Accordingly, we revised the title of our manuscript as follows.

“Molecular Mechanisms Driving Progression of Liver Cirrhosis towards Hepatocellular Carcinoma in Chronic Hepatitis B and C Infections: A Review”

Response to your specific comment 1:Introduction. Since the paper underlines that incidence of HCC in cirrhosis related to HBV and HCV infections is higher in Japan than in Western countries, it would be important to compare epidemiology studies between the two geographical areas (see Fattovich G, Gastroenterology 2004).”

Thank you for your valuable comment.We agree with you. Accordingly, we revised our manuscript as follows.

In page 1, lines 37-42,

HCC is the 4th most common neoplasm and the 2nd commonest cause of cancer deaths in the world. Notably, HCC is a male-dominant disease, with the incidence of HCC ~3-fold higher in males than in females [10].Hepatitis B virus (HBV) infection is associated with the higher HCC incidence in persons with cirrhosis, occurring in high endemic areas and in Western countries (5-year cumulative incidence, 15% and 10%, respectively) [11]. In hepatitis C virus (HCV)-related cirrhosis, the 5-year cumulative HCC risk is 30% in Japan and 17% in Western countries [11].

Response to your specific comment 2:Row 91-93: Age-dependence of the risk to develop HCC must be clarified, taking into account that some comorbidities typically affecting aged patients (i.e. type 2 diabetes) might explain the increased malignant progression of the cirrhotic liver damage.”

Thank you for your valuable comment.We agree with you. Accordingly, we revised our manuscript as follows.

In page 3, lines 94-97,

….was mainly dependent on age and the duration of infection [36]. The evidences are growing that liver steatosis and diabetes mellitus are factors affecting progression to HCC in HCV-infected patients with compensated cirrhosis [11]. These comorbidities typically affecting aged patients might explain the increased malignant progression of the cirrhotic liver damage.Ongoing alcohol….

Response to your specific comment 3:Row 107-108: changes in the risk of HCC development following HCV eradication with DAAs is a controversial and timely issue that must be underlined (see Alberti A, Liver Int 2017).”

Thank you for your valuable comment.We agree with you. Accordingly, we revised our manuscript as follows.

In page 3, lines 112-114,

…. infected with HCV-related liver cirrhosis [48] although changes in the risk of HCC development following HCV eradication with direct-acting antivirals (DAAs) is a controversial [49-51]. Further studies will be needed in DAA-era.

Response to your specific comment 4:Row 143-159: mechanisms related to perturbations of p53/RB pathway are quite confused, maybe a Figure could be of help to clarify them.”

Thank you for your valuable comment.We agree with you. Accordingly, we made a new Figure 2 in the revised manuscript.

Response to your specific comment 5:Row 224: the concept dealing with Coh type is unclear.”

Thank you for your valuable comment.We agree with you. Accordingly, we revised our manuscript as follows.

In page 8, line 234,

….chromosomes. About half of HBV DNA-cell DNA junctions are located within the so-called cohesive end region that lies between two 11-bp direct repeats (DR1 and DR2) in the HBV genome where transcription and replication of the genome are initiated. All of….

Response to your specific comment 6:Row 238-239: given the relevance of HBX in hepatocarcinogenesis, this observation should be expanded.”

Thank you for your valuable comment.We agree with you. Accordingly, we revised our manuscript as follows.

In page 8, lines 247-252,

HBx expression might play a role in hepatocarcinogenesis by interfering with telomerase activity during hepatocyte proliferation [78], which is supported by breakpoint analysis of the HBV genome [75]. HBx functions as transcriptional activator and suppressor and has effects on hepatocellular apoptosis [79,80]. HBx proteins may upregulate transcriptional activation of human telomerase transcriptase [81]. Cis-activation of human TERT mRNA by HBx gene may also be the mechanism in hepatocarcinogenesis [82].

Response to your specific comment 7:Although the review aims at discussing the molecular mechanisms driving progression towards HCC in cirrhosis, the authors draw much attention on HBV infection. Thus, the risk of HCC in HBV occult should be commented.”

Thank you for your valuable comment.We agree with you. Accordingly, we revised our manuscript as follows.

In page 8, lines 257-258,

….the occurrence of HCC [75]. Thus, it is possible that occult HBV infection may also accelerate hepatocarcinogenesis in HBsAg-negative patients to some extent [84].

Response to your specific comment 8:Iconography is limited and the sequence of Figure is not well integrated into the outlines of the main text. The reviewer suggests a swap between the two figures, moving Figure 1 to page 13 before heading #7, whilst Figure 2 would be better appreciated before the heading #4, to sum up the mechanisms involved.”

Thank you for your valuable comment.We agree with you. Accordingly, we revised Figures 1-3 and their locations.

Response to your minor comment:Row 102: it seems to be HCV-related rather than HBV.”

Thank you for your valuable comment.We agree with you. Accordingly, we revised our manuscript as follows.

In page 3, line 106,

Annual rates of HCC occurrence in patients with HCV-relatedcirrhosis are ~4.5% [1].

Reviewer 2 Report

The review paper is well written and would be of interest to clinicians and researchers in hepatology and oncology. There are several minor problems to be solved.

Specific issues:

Figure 1 - Does the hepatic inflammation really increase throughout the progression of disease? I think inflammation can plateau or decreased to some extent during the transition from chronic hepatitis to liver cirrhosis. Please make sure the scheme is appropriately drawn.

Minor points:

L37-38 - "with >700,000 people contracting HCC and ~600,000 HCC-related deaths worldwide" Describe the duration of these numbers (e.g. each year). Same case is seen in L62-63.

L52 - molecular mechanisms underlying the progression of liver cirrhosis to HCC

L144-150 - All sentences were set in italic.

Figure 2 - Replace the figure with the one with more clear letters (higher quality of image).

Author Response

To Reviewer #2: Thank you for your valuable comments and criticisms.

Response to your comment (Specific issues):Figure 1 - Does the hepatic inflammation really increase throughout the progression of disease? I think inflammation can plateau or decreased to some extent during the transition from chronic hepatitis to liver cirrhosis. Please make sure the scheme is appropriately drawn.”

Thank you for your valuable comment.We agree with you. Accordingly, we revised and made new Figure 1.

Response to your comment (minor point 1):L37-38 - "with >700,000 people contracting HCC and ~600,000 HCC-related deaths worldwide" Describe the duration of these numbers (e.g. each year). Same case is seen in L62-63.”

Thank you for your valuable comment.We agree with you. Accordingly, we deleted “L37-38 - "with >700,000 people contracting HCC and ~600,000 HCC-related deaths worldwide"” Concerning “L62-63”, we put “in 2010” in the sentence.

Response to your comment (minor point 2): “L52 - molecular mechanisms underlying the progression of liver cirrhosis to HCC”

Thank you for your valuable comment.We agree with you. Accordingly, we revised our manuscript as follows.

In page 2, lines 55-56 of the revised manuscript,

 “In this review, we discuss the molecular mechanisms underlying the progression of liver cirrhosis to HCC.We expect….”

Response to your comment (minor point 3): “L144-150 - All sentences were set in italic.”

Thank you for your valuable comment. We revised our manuscript accordingly.

Response to your comment (minor point 4): “Replace the figure with the one with more clear letters (higher quality of image).”

Thank you for your valuable comment. We revised figures of our manuscript.

Round  2

Reviewer 1 Report

The authors have addressed the previous issues and the manuscript looks improved.